# Prevalence and Predictors of COVID-19 Vaccination Acceptance among Greek Health Care Workers and Administrative Officers of Primary Health Care Centers: A Nationwide Study Indicating Aspects for a Role Model

**DOI:** 10.3390/vaccines10050765

**Published:** 2022-05-12

**Authors:** Ioanna Avakian, Lemonia Anagnostopoulos, George Rachiotis, Konstantinos Fotiadis, Anargyros Mariolis, Michalis Koureas, Katerina Dadouli, Christos Papadopoulos, Matthaios Speletas, Maria Bakola, Panagiota Vardaka, Stamatia Zoubounelli, Evangelos Tatsios, Fevronia Niavi, Apostolia Pouliou, Christos Hadjichristodoulou, Varvara A. Mouchtouri

**Affiliations:** 1Laboratory of Hygiene and Epidemiology, Faculty of Medicine, University of Thessaly, 22 Papakyriazi Street, 41222 Larissa, Greece; joavakian@uth.gr (I.A.); lanagnost@uth.gr (L.A.); grachiotis@gmail.com (G.R.); mkoureas@uth.gr (M.K.); adadouli@uth.gr (K.D.); xhatzi@med.uth.gr (C.H.); 2Hmathia General Hospital, Veria Hospital Unit, 59132 Veria, Greece; kostasfotiad@yahoo.gr; 3Primary Care Health Center of Areopolis, Areopoli, 23062 Mani, Greece; amariolis@gmail.com; 4Directorate of Public Health, Regional Unit of Kavala, 69132 Kavala, Greece; xpap@pamth.gov.gr; 5Department of Immunology and Histocompatibility, Faculty of Medicine, University of Thessaly, 41110 Larissa, Greece; maspel@med.uth.gr; 6Research Unit for General Medicine and Primary Health Care, Faculty of Medicine, School of Health Science, University of Ioannina, 45110 Ioannina, Greece; m.bakola@uoi.gr; 7Department of General Practice, University Hospital of Larissa, 41334 Larissa, Greece; yiouli_var@windowslive.com (P.V.); matinazoub@gmail.com (S.Z.); evangelos.tats@gmail.com (E.T.); lpouliou@windowslive.com (A.P.); 8Primary Health Care Center of Palama, 43200 Palamas, Greece; fedraniavi@yahoo.gr

**Keywords:** COVID-19, SARS-CoV-2, vaccine, vaccination, health care workers, acceptance, hesitancy, vaccine safety

## Abstract

Background: Τhe study aims to identify factors associated with COVID-19 vaccine acceptance and to investigate knowledge and perceptions of Primary Health Care Centers (PHCC) personnel, who acted as pioneers in the national COVID-19 vaccination strategy. Methods and Materials: A nationwide cross-sectional survey was conducted by distributing an online anonymous questionnaire comprising 25 questions during the first semester of 2021. Results: Approximately 85.3% of the 1136 respondents (response rate 28.4%) were vaccinated or intended to be. The acceptance of seasonal flu vaccine (aOR: 3.29, 95%CI: 2.08–5.20), correct COVID-19 vaccine knowledge (aOR: 8.37, 95%CI: 4.81–14.59) and lack of concern regarding vaccine novelty (aOR: 6.18, 95%CI: 3.91–9.77) were positively correlated with vaccine acceptance. Vaccinated respondents were more likely to be physicians (aOR: 2.29, 95%CI: 1.03–5.09) or administrative staff (aOR: 2.65, 95%CI: 1.18–5.97) compared to nursing stuff. Reasons for vaccine hesitancy included inadequate information (37.8%) and vaccine safety (31.9%). Vaccine acceptance was strongly correlated (Spearman’s correlation coefficient r = 0.991, *p* < 0.001) between PHCC personnel and the general population of each health district. Conclusions: COVID-19 vaccine acceptance among PHCC personnel in Greece was comparably high, but specific groups (nurses) were hesitant. As the survey’s target population could serve as a role model for the community, efforts should be made to improve COVID-19 vaccine acceptance.

## 1. Introduction

After the declaration of a public health emergency of international concern (PHEIC), the World Health Organization (WHO) characterized COVID-19 as a pandemic in March 2020 [1]. By April 2021 and following evaluation by the European Medicines Agency (EMA), four vaccines were authorized for use in the European Union (EU) under conditional marketing authorization [2], in an effort to confront the devastating impacts of the pandemic. Currently, six WHO authorized vaccines are available worldwide [3]. Up until December 2021, more than 270,500,000 cases of COVID-19 were reported globally resulting in over 5,000,000 deaths, while more than 8 billion doses of authorized vaccines have been distributed [4]. In the EU alone, almost 27 million COVID-19 cases and more than 300,000 deaths were reported by December 2021, while over 300,000,000 adults received at least one vaccine dose [4]. Meanwhile, data from the Hellenic National Public Health Organization (NPHO) revealed that in Greece a total of 1,017,445 COVID-19 cases were reported by 15 December 2021, with 19,553 related deaths [5]. The one-dose vaccine coverage for adults in the country reached 69.1% as of 15 December 2021 [6].

WHO has defined vaccine hesitancy as a behavior related to a variety of factors, including confidence, complacency (perceived risks of vaccine-preventable diseases are low) and convenience (access issues) [7]. Previous experience related to the administration of new vaccines, such as the influenza A (H1N1) 2009 vaccine, demonstrated that vaccination hesitancy prevented the achievement of efficient vaccine coverage even among health care personnel, despite the authorities’ recommendations [8,9]. Moreover, WHO highlights that health care sector personnel can play an important role in the successful implementation of a vaccination policy, as they can contribute to vaccination promotion and act as role models for the community [3]. Recent studies revealed that doctors’ advice for COVID-19 vaccination counts in favor of vaccine acceptance [10,11,12]. In Greece, the current COVID-19 vaccination project involves all personnel working in Primary Health Care Centers (PHCC) [13]. PHCC are publicly funded and consist of Rural Health Centers (RHC) and Urban Health Units (UHU). PHCC staff include Health Care Workers (HCW), such as physicians, nurses and other health care professionals (social and welfare, health promotion, midwives, ambulance personnel, etc.), and Administrative Officers (AO). PHCC responsibilities involve medical care, health promotion, vaccinations and health consultations (breast feeding, mental health, actions in schools and community centers), all of which build a strong bond between PHCC professionals and civilians.

Although the vaccination coverage of Greek health care personnel is continuously monitored by the NPHO and communicated to the European Centre for Disease Prevention and Control (ECDC) [6], information about subgroups of HCWs and AOs is not formally available. Surveys conducted among the general population regarding attitudes toward future vaccination have shown that, early in the pandemic (April–May 2020), 18.9% of respondents declared they were against vaccination, while 81.1% declared that they may consider or will be vaccinated [14]. Since PHCC personnel play an essential role in Greece for the provision of information and vaccine administration, data related that their COVID-19 vaccine acceptance may contribute to the assessment and improvement of the Greek vaccine campaign, by focusing on the above target group if necessary. In late July 2021, the Greek government introduced legislation on the mandatory COVID-19 vaccination of personnel of health care facilities in both the public and private sectors. However, up until April 2022, the vaccination coverage of HCW, according to ECDC COVID-19 vaccines tracker, reached 90.7%, despite the legislation requirement for mandatory vaccination [15]. Approximately 9% of HCW preferred to abandon their job or to have unpaid leave for several months, instead of getting the COVID-19 vaccine. Data related to COVID-19 vaccine acceptance/hesitancy is important, in order to understand the profile of hesitant professionals and adapting vaccination policies accordingly. The aim of our study is to: (a) estimate both the intention and uptake of COVID-19 vaccination among PHCC personnel, (b) identify factors related to their decision to get vaccinated, (c) investigate perceptions and behavioral aspects of PHCC personnel in relation to vaccination and (d) identify aspects that can facilitate PHCC personnel to be role models for the general population.

## 2. Methods and Materials

### 2.1. Study Design

A cross-sectional online questionnaire-based survey was conducted between February and June 2021 when authorized COVID-19 vaccines were available in Greece, in order to assess the knowledge, attitudes and practices of PHCC personnel (HCWs and AO), as well as factors related to COVID-19 vaccine acceptance and hesitancy. A sample size of 916 was calculated using a Raosoft Digital Sample Size Calculator [16], in which 3% was used as a margin of error, 95% as the confidence interval (CI), 50% as the expected frequency and 6456 as the population size [17]. Due to the online distribution format, the estimated response rate was approximately 22%, and a sample of 4000 was calculated [18]. A geographically stratified sampling plan based on Greek health districts was applied to produce a representative sample of 125 PHCC, located both in mainland and Greek islands. 

An anonymous online questionnaire consisting of 25 questions was designed after considering guidelines published by WHO, advice issued by the Hellenic National Public Health Organization (NPHO) and the Ministry of Health, as well as relevant studies conducted in Greece [9,14,18,19,20]. The study was supervised by an expert team comprising an epidemiologist, an occupational health professional and a public health specialist who were responsible for the face and content validation of the questionnaire. The pilot testing of the first draft questionnaire was conducted to evaluate the time required for its completion, to appraise the clarity of the questions addressed to professionals from various backgrounds and to test the online tool functionality. A total of 20 PHCC professionals, including HCWs and AOs, completed the draft questionnaire, which was modified appropriately to produce the final version of the survey. Questionnaires completed during pilot testing were excluded from the final analysis. Internal consistency and reliability of the questionnaire was assessed by estimating an Cronbach’s alpha value of 0.70, which was considered as acceptable [21].

The questionnaire included questions about knowledge, attitude and practice (KAP) regarding vaccinations in general as well as the COVID-19 vaccination specifically (Appendix A). The time required for completion of the questionnaire was approximately 15 min. The collection, entry, analysis and storage of survey data complied with the anonymity, privacy and confidentiality regulations of the national legislation and rules of the University of Thessaly, Greece. In order to identify whether COVID-19-vaccination-hesitant professionals were less acceptant of vaccination generally, the survey questionnaire included questions related to knowledge, attitudes and practices regarding general vaccination (questions 14–17) and questions related to COVID-19 vaccines specifically (question 22).

The researchers contacted selected PHCC to distribute the questionnaire, which was forwarded through email to PHCC personnel. The email contained an invitation letter detailing the survey protocol, issues regarding confidentiality and the researchers’ contact information. Moreover, the cover letter provided a link to the online questionnaire and emphasized voluntary participation. Electronic and hard copies of the cover letter and questionnaire were distributed, with reminders sent to increase the response rate [18,22]. Participants had the option of completing either the electronic or hard copy after written consent was obtained. Online answers were stored automatically, while the hard copy questionnaires were sent by courier to researchers. Data were entered in the database when needed from hard copies by trained staff. PHCC emails and contact details were obtained from the Regional Health Authorities’ websites.

From 11 February (approximately one month after the initiation of COVID-19 vaccination) to 30 June 2021, the structured and anonymous questionnaire was distributed to PHCC, with both HCWs and AOs invited to participate. The questionnaire (Appendix A) included 11 questions related to demographics and 14 questions for the assessment of knowledge, attitudes and practices concerning immunization generally and for COVID-19 specifically. The general section of the questionnaire included questions about demographics, education, workplace and length of work experience. Part A included generic questions related to vaccination, while part B referred specifically to COVID-19 and vaccination. In part A, the first two questions referred to respondents and their family vulnerability, according to their medical history (answer: Yes/No). This was followed by questions about knowledge and attitudes/perceptions. Each category had three sub-questions (three for knowledge and three for attitudes). Answers were given on a 5-level item scale (“completely agree”, “agree”, “neither agree nor disagree”, “disagree” and “completely disagree”). The last two questions in part A applied to vaccination practices, either to respondents’ children if any, or to themselves regarding the seasonal influenza vaccine. Those who responded negatively about seasonal influenza vaccination were asked to explain their response through a semi-closed question. In part B, respondents were asked about any contact they had with a COVID-19 patient during either social or professional activities (Yes/No). A question regarding self-evaluation of their COVID-19 knowledge followed (four-level item: Non-existent/Insufficient/Satisfactory/Excellent) and another question related to their source of information. COVID-19 vaccine knowledge was evaluated through three questions, which were answered using a 5-level item scale (“completely agree”, “agree”, “neither agree nor disagree”, “disagree” and “completely disagree”). COVID-19 vaccine acceptance was measured through the question “Have you or will you be vaccinated with one of the vaccines against SARS-CoV-2 virus that causes COVID-19, which has been approved by the National Pharmaceutical Organization?”. Participants who responded negatively were requested to specify the reason (semi-closed question). Finally, all participants were asked if the short period for COVID-19 vaccine development (vaccine novelty) concerned them and if they believed in mandatory vaccination for health care professionals (Yes/No).

### 2.2. Ethical Statement

The questionnaire was approved by the Ministry of Health and the Regional Health Authorities. Ethical approval from the scientific committee of the University of Thessaly (protocol number 49/13 January 2021) was obtained. All participants provided written consent before completing the questionnaire.

### 2.3. Statistical Analysis

Data were analyzed using IBM SPSS Statistics for Windows, Version 25.0 (IBM Corp., Armonk, NY, USA). Continuous variables were expressed as means ± standard deviations, and categorical variables as frequencies and percentages. The relationship between the main outcome measure (acceptance of the COVID-19 vaccine) and participants’ characteristics (baseline characteristics, perception and knowledge about the COVID-19 vaccine) were assessed using either chi-square analysis or Student’s *t*-test. A Student’s *t*-test was performed for continuous data since there was no deviation from normal distribution (Shapiro–Wilk normality test) and violation of the assumption of homogeneity of variance (Levene’s test). In univariate analysis, the percentage of those vaccinated and the proportional ratio (PR) with 95% confidence intervals (CIs) were presented. The direction of the association was analyzed using a bivariate logistic regression analysis with a 95%CI. The selection of variables for the bivariate logistic regression model were based on factors previously reported in the literature and found to be significant in the chi-square analysis or Student’s *t*-test. Spearman’s correlation coefficient was used to measure the strength and direction of association between the percentage of vaccinated PHCC personnel and percentage of vaccinated adults (at least one dose) in each health district. Population data for each health district/prefecture were obtained from the Hellenic Statistical Authority (ELSTAT)and the National Vaccination Registry [23,24]. All tests were 2-sided and a *p*-value of <0.05 was considered to indicate statistical significance.

Certain survey questions (questions 14, 15, 22 and 24) were rated on a 5-point scale as follows: “completely disagree”, “disagree”, “neither agree nor disagree, agree” and “completely agree”. The responses “completely disagree”, “disagree” or “neither agree nor disagree” were considered to indicate disagreement, while responses of “completely agree” or “agree” were taken as agreement. Survey questions 14, 15 and 22 each consisted of three sub-questions. Correct answers to all three sub-questions were considered as a correct answer, whereas answering at least one of the three sub-questions incorrectly was considered as an incorrect answer.

Regarding sources of information, through univariate analysis, two groups were created. The first group included formal sources of information (medical articles in journals, committee for infectious diseases at the health facility, websites of the NPHO and the Hellenic Ministry of Health), while the other group included informal information sources (television, social media channels, newspapers and general interest publications/journals/websites). Each source of information counted a frequency score up to 4 depending on the participant’s answer (1 = always, 2 = often, 3 = rarely, 4 = never). The analysis was based on the relevant frequency score in order to categorize respondents to each group.

## 3. Results

### 3.1. Basic Demographics

A total of 4000 questionnaires were disseminated to PHCC personnel, with 1136 HCWs and AOs having responded (response rate: 28.4%). The majority of participants were female (69.8%), married (69.1%) and the average age was 43.8 years; most participants possessed a bachelor’s or a master’s degree (63.4%). Participants’ occupations covered the entire spectrum of primary care, including nursing staff (25.8%), physicians (46.1%) and other health care workers (social/welfare workers, midwives, health promotion specialists (10.1%); other health care workers (laboratory staff, radiologists and ambulance crew) (9.2%); and administrative officers (8.8%)). The vast majority of the respondents worked in a RHC (86.5%), and the median years of practice was 15 (IQR = 10). The percentage of vulnerable participants due to their medical history and participants living with vulnerable people were 17.2% and 27.9%, respectively. Questions 14 and 15, which were related to generic knowledge and perceptions towards vaccination, were answered correctly by 70.6% and 45.7% of respondents, respectively. More than half of respondents (61.2%) declared that they were parents, and of these respondents 98.1% adhered to the National Vaccination Program. The majority of respondents declared that they were vaccinated regularly for seasonal influenza (79%). Most participants knew individuals who experienced COVID-19 infection (85.1%), while 75.6% of respondents were contacts of a COVID-19 patient in the workplace. Most respondents rated the information they obtained related to COVID-19 as “excellent” of “satisfactory” (15.3% and 62%, respectively), while 1.1% believed that they were “uninformed”. More than half of the participants (57.9%) answered questions related to knowledge about COVID-19 vaccines correctly. From 1136 participants, 969 (85.3%) declared that they were fully vaccinated or intended to receive a COVID-19 vaccine. Finally, 68.9% of respondents were concerned about the novelty of the vaccine and its rapid development, while 53.1% agreed with a vaccine mandate for health care professionals.

### 3.2. Univariate and Multivariate Analysis

The results of univariate analysis are presented in Table 1, Table 2 and Table 3.

Factors that were positively associated with vaccine acceptance emerged through univariate analysis and included a higher education level (PR: 1.26, 95%CI: 1.13–1.41), being a physician (PR: 1.22, 95%CI: 1.14–1.31), working in a specific health district (PR: 1.19, 95%CI: 1.14–1.25), a higher score on vaccination knowledge (Q14) and perceptions (Q15) (PR: 1.24 95%CI: 1.16–1.33 and PR: 1.24, 95%CI: 1.18–1.30, respectively), accepting influenza vaccination (PR: 1.44 95%CI, 1.30–1.58) and adhering to the National Child Vaccination Program as a parent (PR: 1.71, 95%CI: 0.97–3.02). 

Table 4 describes the results from the multivariate analysis.

A statistically significant association was found among COVID-19 vaccination and specific occupations, health district of employment, and being vaccinated for seasonal influenza (aOR: 3.29, 95%CI: 2.08–5.20). Moreover, correctly answering questions related to knowledge about COVID-19 vaccines (aOR: 8.37 95%CI: 4.81–14.59) and fewer concerns about the novelty of vaccines and their rapid development (aOR: 6.18, 95%CI: 3.91–9.77) were both positively associated with vaccine acceptance.

Figure 1 demonstrates the reasons for COVID-19 vaccination refusal among study participants. 

According to data from the National Vaccination Registry, at the time of our survey, the countrywide proportion of COVID-19 vaccination in the general adult population was approximately 42% [24]. A strong positive correlation was identified between the percentage of vaccinated PHCC personnel and vaccinated adults in the general population (at least one dose) in each health district (Spearman’s correlation coefficient ρ = 0.991, *p* < 0.001) (Figure 2 and Table 5).

Further analysis was conducted to investigate the association between knowledge/perception questions and sources of information, demographic characteristics and determinants of COVID-19 vaccination acceptance. The results of the aforementioned analysis are presented in Appendix A. Regarding the association between information sources and COVID-19 vaccination acceptance, formal sources of information about COVID-19 vaccination were positively associated with COVID-19 vaccine acceptance: articles in scientific medical journals (PR: 1.15, 95%CI: 1.08–1.23), infection control committee at a health facility (PR: 1.09, 95%CI: 1.04–1.15), NPHO website (PR: 1.19, 95%CI: 1.10–1.29) and the Hellenic Ministry of Health website (PR: 1.11, 95%CI: 1.05–1.18) (Appendix A.)

Variables associated with general vaccination knowledge (Q14) and positive vaccination perceptions (Q15) as well as COVID-19 vaccine knowledge (Q22) are shown in Appendix A. Generally, the top score to the aforementioned questions was positively associate with a higher educational level (PR: 1.98, 95%CI: 1.60–2.46, PR: 2.83, 95%CI: 1.96–4.09, PR: 2.52, 95%CI: 1.87–3.38), being a physician or health promotion specialist (PR: 1.49, 95%CI: 1.37–1.63, PR: 2.40, 95%CI: 2.04–2.82, PR: 1.61, 95%CI: 1.44–1.81), being vaccinated against seasonal influenza (PR: 1.26, 95%CI: 1.16–1.37, PR: 1.31, 95%CI: 1.23–1.39, PR: 1.24, 95%CI: 1.16–1.33), having less concerns about COVID-19 vaccine novelty (PR: 1.3, 95%CI: 1.17–1.44, PR: 1.45, 95%CI: 1.34–1.57, PR: 1.37, 95%CI: 1.25–1.49) and believing in the COVID-19 vaccine mandate (PR: 1.3, 95%CI: 1.17–1.44, PR: 1.37, CI: 1.23–1.53, PR: 1.34, 95%CI: 1.19–1.51).

### 3.3. Internal Consistency Reliability

The internal consistency of the questionnaire was established by calculating Cronbach’s alpha coefficient. The reliability coefficient was calculated at 0.732, suggesting an acceptable internal consistency.

## 4. Discussion

Our study provided an appraisal of COVID-19 vaccination acceptance among PHCC personnel in Greece (HCWs and AOs) and of factors related to the decision making of health care personnel during the first months in which they were able to access COVID-19 vaccines. During the study period, the vaccination of both HCW and the general public was optional. The target group of our study was expected to play a major role in promoting COVID-19 vaccination to the general population and data concerning their intention to get vaccinated and factors related to their decision were important to be investigated when designing and implementing the vaccination strategy. Vaccination monitoring among health care personnel has shown that approximately 9% of HCW refused the COVID-19 vaccination as of April 2022 [15]. Our study reported an 85.3% vaccine acceptance during the study period (from February to June 2021), while data available from the ECDC COVID-19 vaccines tracker reported that the total vaccination coverage among HCW in Greece is currently 90.7%, which demonstrates an increase of approximately 5%, after the introduction of legislation for the mandatory COVID-19 vaccination of HCW since late July 2021 [15]. Unfortunately, no follow up study was conducted by our team. An ongoing international survey coordinated by WHO is expected to give more insights on changes in vaccination coverage over time and identify factors related to vaccine hesitancy [25].

Vaccine acceptance in our study (85.3%) was found to be higher compared to other studies conducted in Greece. In particular, three studies conducted before the availability of an effective vaccine showed vaccine acceptance to be 51.1% [26], 78.5% [19] and 43% [20]. However, one study conducted after the release of COVID-19 vaccines demonstrated compatible findings (an acceptance prevalence of 85.3%), though this study’s target population included only physicians [27]. During the same period, our research group conducted a similar study (similar questionnaire, different mode of distribution and different target group) among personnel working in Greek hospitals and providing secondary health care services; the vaccine acceptance prevalence among the survey population was 77.7% [28]. The higher level of vaccination acceptance among our study group may be reflective of the primary health care personnel’s greater involvement in the Greek vaccination program, and may have been influenced by the questionnaire’s mode of distribution (online versus paper based). The estimated prevalence of vaccine acceptance in our study was higher than the actual vaccine coverage recorded by the Ministry of Health’s National Registry and communicated to ECDC during the study period (77% at least one dose and 70.8% fully vaccinated) [29]. However, our study estimates both the intention and attainment of vaccination, in contrast to the National Registry data, which show actual vaccinations. 

Regarding the global situation, studies conducted prior to vaccination implementation presented generally lower acceptance rates [30,31,32,33,34], even when the studies’ target groups included professionals responsible for national vaccination project implementation [35]. However, a study in South Africa demonstrated a vaccine acceptance of 90.1% [36]. Many studies conducted, for example, in Germany [37], Canada [38], India [39] and the USA [40] following the release of vaccines describe a vaccine acceptance level comparable with our study. However, other studies conducted in Germany [41], the United Arabic Emirates [42], Czech Republic [43] and the USA [44] reveal lower vaccine acceptance levels compared to our study.

In our survey, no association with age, sex and gender variables was detected. However, several studies reported a higher vaccine acceptance by males and individuals of older age [34,38,43,45,46]. One Italian study supported younger age as a predictor of COVID-19 vaccine acceptance [33].

Several surveys reported a lower COVID-19 vaccine acceptance among nurses compared to physicians [28,34,38,47], and our study confirms this observation. Nurses’ vaccine hesitancy could be considered as a risk factor for COVID-19 transmission, due to their prolonged and close contact with health facility users. Updating nursing educational curriculum to focus on public health issues and in particular vaccination as one of the most important measures to combat infectious diseases, collective responsibility, risk perception and communicable disease prevention, during both undergraduate and postgraduate studies, could increase nurses’ competencies in health promotion and improve their vaccine acceptance.

The seasonal influenza vaccination of PHCC personnel was one of the predictors for COVID-19 vaccination acceptance. Our results are in line with several similar studies where a strong correlation was observed [11,27,28,33,34,42,43,44,45,48,49].

Our survey revealed that participants who were well informed about COVID-19 vaccines were more likely to accept vaccination. Similar findings were reported by another study conducted in five European countries [11]. However, a better generic knowledge of or positive attitudes towards vaccination in general terms had no association with acceptance or hesitancy. Moreover, major reasons for vaccine hesitancy included inadequate information about COVID-19 vaccines and possible vaccine side effects. Most studies worldwide depict similar results [30,31,35,43,45], which underlines the need for specific and continuous information regarding COVID-19 vaccination, in order to improve relative knowledge and reduce concerns about safety and effectiveness.

According to our study results, participants with fewer concerns about vaccine novelty and the short timeframe for vaccine development were more prone to vaccination. 

Concerning geographical data, certain health districts have lower vaccination rates compared to others. Specifically, participants working in health districts located in northern Greece show a greater level of vaccination hesitancy. The results are in line with a survey conducted by our research group in personnel of Greek hospitals [28]. This observation is significant in the context of a relative national vaccination campaign, as it reveals that national campaigns may have variable impacts in different areas.

In late July 2021, the Greek government introduced legislation regarding the mandatory vaccination for personnel working in health care facilities throughout the country. Our study indicates that more than half of the participants agreed with mandating COVID-19 vaccination for health care professionals. However, vaccine mandate studies are divided, both supporting [28,50] and against [41,51] the mandates. By the end of April 2022, more than 8 months after introducing legislation for the mandatory vaccination of HCW, 9.3% of HCW continue to refuse vaccination, despite financial and professional adverse consequences. The results of our survey depicted that mandatory legislation might not be 100% successful in addressing vaccine hesitancy.

Information sources are expected to be a factor for vaccine acceptance. Our study depicted that formal/national source of information may be facilitators for COVID-19 vaccination. Several surveys indicated that relying on traditional sources of information and governmental guidelines positively affects vaccination acceptance, while information sourced from social media and general context websites encourage vaccine hesitancy [19,27,33,37,52]. Intensive research in this field may be necessary and could improve understanding the dynamics of these findings. 

Another noteworthy result identified is the strong correlation between PHCC personnel COVID-19 vaccine acceptance and adult population vaccination coverage against COVID-19 in each health district. This finding is similar to another survey that our research group conducted among personnel of Greek hospitals [28]. However, the higher level of correlation in the present survey might demonstrate the vast impact of PHCC personnel behavior in local communities and emphasizes the possibility of PHCC personnel acting as opinion leaders for the general population.

Our study has limitations: The sample was convenient although the response rate was acceptable for a hybrid format (online and paper-based) study [22]. Participation (selection) bias may have occurred, as vaccinated personnel may be more willing to participate, despite the anonymity provided through the online questionnaire. At face value, it seems that our study yields a low response rate, and since we were not able to collect data from non-respondents; this may be a source of selection bias. It should be noted that a low response rate to online surveys has been previously reported [53]. Nevertheless, our survey response rate of approximately 30% could be considered as relatively satisfactory, taking into account the online method of data collection employed during the era of COVID-19. Unfortunately, due to heavy workloads of PHCC personnel during the study period, the length of questionnaires should be limited to maximize participation. For convenience, useful information was not included in the questionnaire: Hepatitis B vaccination, recommendation for COVID-19 vaccination to the general population, revealing elements of character such as altruistic and self-serving behavior, reasons for vaccine hesitancy such religion or trust in pharmaceutical/formal authorities [7]. Despite these shortcomings, our study has the considerable advantages of a nationwide nature of HCWs sampled, and the use of a detailed questionnaire on a wide spectrum of knowledge, attitudes and practices of primary health care personnel towards vaccinations. The primary health care workforce could play a pivotal role in the promotion of vaccination coverage, including COVID-19 related vaccines, since primary care staff have a history of experience successfully delivering immunization programs [54]. Moreover, the primary care setting has the potential to combine facets of knowledge, attitudes, behavior, culture and health in the concept and practice of personalized care. In addition, patients feel comfortable receiving medical care from health care practitioners who are culturally compatible in understanding their concerns [55]. Consequently, the primary care environment is the best source of trusted information for those who are hesitant towards vaccination. 

## 5. Conclusions

PHCC personnel appeared to accept the COVID-19 vaccine during the first months of 2021. However, specific occupational groups, such as nurses, demonstrated a greater hesitancy to the vaccine than others. Efforts should be made to combat the objections of hesitant employees and improve their acceptance towards new vaccines, as there are indications of their being a role model for the community. Several cultural and behavioral barriers might play a significant role to professionals denying COVID-19 vaccination, which seem to be present even after mandatory measures have been taken. Additional studies in the field could provide useful information and tools to tackle vaccine hesitancy more efficiently. Adequate and constant information/education about COVID-19 vaccines could be the major tool to increase vaccination coverage of both HCW and general population. Vaccination strategies should consider the reasons that HCW refused COVID-19 vaccines as reported by the responders of our study. Addressing fears and providing further information about vaccines through information campaigns for HCW could potentially increase vaccine coverage among HCW.

## Figures and Tables

**Figure 1 vaccines-10-00765-f001:**
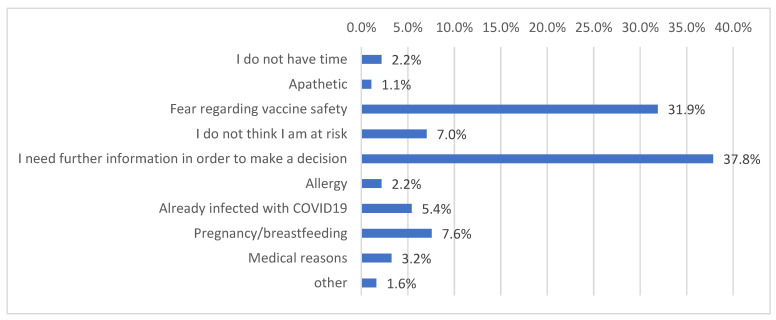
Distribution of reasons for COVID-19 vaccination refusal.

**Figure 2 vaccines-10-00765-f002:**
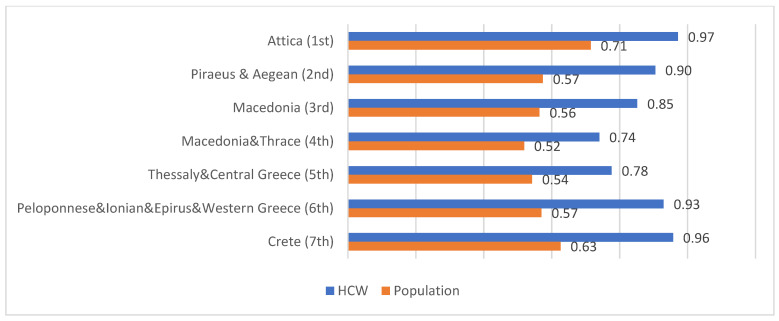
COVID-19 vaccination percentages of PHCC personnel and general population in each health district.

**Table 1 vaccines-10-00765-t001:** Demographic characteristics associated with COVID-19 vaccine acceptance, expressed with proportional ratio (PR) in univariate analysis (Total number of respodents 1136).

Variables		*n* (%)	Vaccinated *n* (%) or Median (IQR)	Proportional Ratio (PR) 95%CI	Sig.
Age		44 (13.0)	Vac: 968, 44 (13.8) Non-vac: 167,44 (14)	-	0.884
Gender	Male	343 (30.2)	296 (86.3)	1.02 (0.97–1.07)	0.584
	Female	793 (69.8)	673 (84.9)	Ref.	
Marital status	Married	755 (69.1)	642 (85.0)	1.00 (0.95–1.06)	0.930
	Divorced	31 (2.8)	27 (87.1)	1.03 (0.89–1.19)	0.999 (F) *
	Widowed	4 (0.4)	4 (100)	1.18 (1.12–1.24)	0.999 (F) *
	Ν/A	43 (3.8)	39 (90.7%)	1.07 (0.96–1.19)	0.363 (F) *
	Unmarried	303 (27.7)	257 (84.8)	Ref.	
Educational level	Master/Doctoral	299 (26.3)	268 (89.6)	1.33 (1.10–1.62)	<0.001
	Higher Education Institute/University (BSc, AΕΙ)	421 (37.1)	382 (90.7)	1.35 (1.11–1.63)	<0.001
	Technological Educational Institute (TΕΙ)	279 (24.6)	224 (80.3)	1.19 (0.98–1.45)	0.037
	Institute of Vocational Training (IEK)	85 (7.5)	60 (70.6)	1.05 (0.83–1.50)	0.686
	High school	52 (4.6)	35 (67.3)	Ref.	
Educational level (groups)	Higher Education Institute/University (AΕΙ) and Technological Educational Institute (TΕΙ) and Master or Doctoral	999 (87.9)	874 (87.5)	1.26 (1.13–1.41)	<0.001
	High school and Institute of Vocational Training (IEK)	137 (12.1)	95 (69.3)	Ref.	
Health care profession	Health promotion specialist	115 (10.1)	101 (87.8)	1.16 (1.06–1.27)	0.007
	Other Health Professionals	104 (9.2)	78 (75.0)	0.99 (0.87–1.13)	0.876
	Administrative	100 (8.8)	82 (82.0)	1.08 (0.97–1.21)	0.199
	Physician	524 (46.1)	486 (92.7)	1.22 (1.14–1.31)	<0.001
	Nursing staff	293 (25.8)	222 (75.8)	Ref.	
Sector of employment	Rural Health Center (RHC)	983 (86.6)	833 (84.7)	0.95 (0.90–1.01)	0.178
	Urban Health Unit (UHU)	153 (13.4)	136 (88.9)	Ref.	
Health district of employment	1st	71 (6.2)	69 (97.2)	1.31 (1.20–1.44)	<0.001
	2nd	84 (7.3)	76 (90.5)	1.22 (1.10–1.36)	0.002
	3rd	182 (16.0)	155 (85.2)	1.15 (1.04–1.28)	0.007
	5th	246 (21.7)	191 (77.6)	1.05 (0.94–1.17)	0.370
	6th	283 (24.9)	263 (92.9)	1.26 (1.15–1.37)	<0.001
	7th	70 (6.2)	67 (95.7)	1.30 (1.18–1.42)	<0.001
	4th	200 (17.6)	148 (74.0)	Ref	
Health district of employment (groups)	3,4,5	628 (55.3)	494 (78.7)	Ref.	
	1,2,6,7	508 (44.7)	475 (93.5)	1.19 (1.14–1.25)	<0.001
Years of practice		14 (17.0)	Vac: 14 (17.0) Non-vac: 14 (18.0)	-	0.969

* Fisher’s exact test.

**Table 2 vaccines-10-00765-t002:** Univariate analysis of the generic knowledge and attitudes towards vaccines and COVID-19 vaccine acceptance (Section A) (N = 1136).

Variables		Yes (%) or Correct (%)	Vaccinated Yes/Correct (%)	Vaccinated No/Incorrect (%)	Proportional Ratio (PR) 95%CI	Sig.
12. Do you belong to a vulnerable/high risk group due to your medical history?		195 (17.2)	162 (83.1)	807 (85.8)	0.97 (0.91–1.04)	0.336
13. Do you live with older individuals or individuals belonging to a vulnerable/high risk group due to their medical history?		317 (27.9)	265 (83.6)	704 (86.0)	0.97 (0.92–1.03)	0.313
14a, 14b, 14c (Generic knowledge about vaccination)		802 (70.6)	726 (90.5)	243 (72.8)	1.24 (1.16–1.33)	<0.001
14a.The HPV vaccine is recommended for all males up to 18 years of age in the country		324 (28.5)/	291 (89.8)	678 (83.5)	1.08 (1.03–1.13)	0.007
14b. After the flu vaccination, certain foods are not permitted to be consumed for a period of 24 h		672 (59.2)	619 (92.1)	350 (75.4)	1.22 (1.15–1.29)	<0.001
14c. One of the contraindications of the flu vaccine is an allergy to eggs		587 (51.7)	536 (91.3)	433 (78.9)	1.16 (1.10–1.22)	<0.001
15a, 15b, 15c (Attitudes towards vaccination)		519 (45.7)	495 (95.4)	474 (76.8)	1.24 (1.18–1.30)	<0.001
15a Vaccinations are an important tool for the protection of public health, and in particular for health professionals and workers in the health sector		1080 (95.1)	944 (87.4)	25 (44.6)	1.96 (1.46–2.63)	<0.001
15b. Natural immunity acquired via disease is always preferable to immunity acquired via vaccination		695 (61.2)	654 (94.1)	315 (71.4)	1.32 (1.24–1.40)	<0.001
15c. Many vaccines often have serious side effects		673 (59.3)	621 (92.3)	348 (75.2)	1.23 (1.16–1.30)	<0.001
17. Have you been vaccinated with the seasonal flu vaccine?		898 (79.0)	818 (91.1)	151 (63.4)	1.44 (1.30–1.58)	<0.001
* 16. Are you the parent/guardian of one or more children?		695 (61.2)	588 (84.6)	381 (86.4)	0.98 (0.93–1.03)	0.406
** Do you adhere to the child vaccination program suggested by the National Vaccination Program in the country? (total 689)	YES, I vaccinate my children according to the National Vaccination Program	676 (98.1)	579 (85.7)		1.71 (0.97–3.02)	0.001
	I select and carry out some vaccinations	12 (1.7)	6 (50.0%)		Ref.	
	I do not vaccinate my children	1 (0.1)				

* Of 1136 respondents, 441 did not have a child; ** Of the 695 individuals who had a child, 689 (99,1%) responded.

**Table 3 vaccines-10-00765-t003:** Univariate analysis of the knowledge and attitudes towards COVID-19 vaccines and COVID-19 vaccine acceptance (Section B) (N = 1136).

Variables	Yes (%) or Correct (%) or Disagree (%)	Vaccinated Yes/Correct (%)	Vaccinated No/Incorrect (%)	Proportional Ratio (PR) 95%CI	Sig.
18. Do you know of a relative or friend who has had COVID-19?	967 (85.1)	830 (85.8)	139 (82.2)	1.04 (0.97–1.12)	0.239
19. Do you come into contact with COVID-19 patients while performing your job duties?	859 (75.6)	737 (85.8)	227 (82.2)	1.05 (0.99–1.12)	0.070
22a,22b, 22c (Knowledge about COVID-19 vaccines)	658 (57.9)	635 (96.5)	334 (69.9)	1.38 (1.30–1.47)	<0.001
22a. Some of the vaccines against SARS-CoV-2 which are approved and used in the country are based on mRNA technology	1048 (92.3)	913 (87.1)	56 (63.6)	1.37 (1.17–1.61)	<0.001
22b. The dosage regimen of the vaccines against SARS-CoV-2 includes 3 doses	844 (74.3)	749 (88.7)	220 (75.3)	1.18 (1.10–1.26)	<0.001
22c. There is evidence that mRNA technology interferes with the DNA of cells	792 (69.7)	754 (95.2)	215 (62.5)	1.52 (1.40–1.66)	<0.001
24. Does the short period of time for development of the vaccines cause you any concerns about its safety?	783 (68.9)	742 (94.8)	227 (64.3)	1.47 (1.36–1.60)	<0.001
25. Do you believe that vaccination against SARS-CoV-2 should be mandatory for health care professionals?	603 (53.1)	583 (96.7)	386 (72.4)	1.34 (1.26–1.41)	<0.001
**Variables**		***n* (%)**	**Vaccinated (%)**	**Proportional Ratio (PR) 95%CI**	**Sig.**
20. How do you evaluate your level of being informed about vaccines against the SARS-CoV-2 virus that causes COVID-19?	Excellent	174 (15.3)	165 (94.8)	2.06 (1.14–3.70)	<0.001
	Satisfactory	710 (62.5)	626 (88.2)	1.91 (1.06–3.43)	<0.001
	Insufficient	239 (21.1)	172 (72.0)	1.56 (0.86–2.82)	0.061
	No information	13 (1.1)	6 (46.2)	Ref.	
** 21. Which channels do you use to keep informed about the COVID-19 pandemic and the SARS-CoV-2 vaccine, and how often?	Medical articles in journals, committee for infectious diseases at the health facility, website of the Hellenic National Public Health Organization (NPHO), website of the Hellenic Ministry of Health	702 (61.8)	602 (85.8)	1.01 (0.96–1.06)	0.581
	Television, social media channels, newspaper, general interest publications/journals/websites	434 (38.2)	367 (84.6)	Ref.	

** Analysis using a score related to the frequency with which participants used the information sources of the two groups: “Medical articles in journals, committee for infectious diseases at the health facility, website of the Hellenic National Public Health Organization (NPHO), website of the Hellenic Ministry of Health” vs. “Television, social media channels, newspaper, general interest publications/journals/websites” (see Section 2).

**Table 4 vaccines-10-00765-t004:** Factors associated with COVID-19 vaccine acceptance, expressed with adjusted odds ratio (aOR), in a multivariable analysis.

		Multivariate		
		aOR 95%CI		Sig.
Age		1.02	0.98–1.06	0.279
Gender (Male/Female)		0.60	0.35–1.01	0.054
Educational level	High School	Ref.		
	IEK	0.83	0.32–2.16	0.699
	TEI	1.50	0.65–3.49	0.345
	BSc	0.75	0.27–2.13	0.590
	MSc, PhD	0.68	0.25–1.81	0.439
Health care Profession	Nursing staff	Ref.		
	Health promotion specialist	1.08	0.48–2.43	0.860
	Other Health Professionals	1.38	0.67–2.87	0.386
	Administrative	2.65	1.18–5.97	0.018
	Physician	2.29	1.03–5.09	0.042
Health district of employment (Υ.ΠΕ) (3,4,5)/(1,2,6,7)		0.29	0.17–0.48	<0.001
Years of practice		1.00	0.97–1.04	0.836
12. Do you belong to a vulnerable/high risk group due to your medical history? (yes/no)		0.84	0.48–1.48	0.548
13. Do you live with older individuals or individuals belonging to a vulnerable/high risk group due to their medical history? (yes/no)		0.93	0.57–1.50	0.754
14a, 14b, 14c (correct/incorrect) *		1.30	0.82–2.05	0.260
15a, 15b, 15c (correct/incorrect) **		1.60	0.89–2.86	0.115
17. Have you been vaccinated with the seasonal flu vaccine? (yes/no)		3.29	2.08–5.20	<0.001
18. Do you know of a relative or friend who has had COVID-19? (yes/no)		1.03	0.56–1.90	0.918
19. Do you come into contact with COVID-19 patients while performing your job duties? (yes/no)		1.41	0.83–2.40	0.205
22a, 22b, 22c (correct/incorrect) ***		8.37	4.81–14.59	<0.001
24. Does the short period of time for development of the vaccines cause you any concerns about its safety? (Disagree/agree)		6.18	3.91–9.77	<0.001
aOR = adjusted odds ratio				

* 14a.”The HPV vaccine is recommended for all males up to 18 years of age in the country”; 14b. “After the flu vaccination, certain foods are not permitted to be consumed for a period of 24 h”; 14c. “One of the contraindications of the flu vaccine is an allergy to eggs.” ** 15a “Vaccinations are an important tool for the protection of public health and in particular for health professionals and workers in the health sector”; 15b. “Natural immunity acquired via disease is always preferable to immunity acquired via vaccination”; 15c. “Many vaccines often have serious side effects”. *** 22a. “Some of the vaccines against SARS-CoV-2 which are approved and used in the country are based on mRNA technology”; 22b. “The dosage regimen of the vaccines against SARS-CoV-2 includes 3 doses”; 22c. “There is evidence that mRNA technology interferes with the DNA of cells”.

**Table 5 vaccines-10-00765-t005:** Correlation between PHCC personnel COVID-19 vaccine acceptance and adult population vaccination coverage against COVID-19.

			HCW	Pop
Spearman’s rho	HCW	Correlation Coefficient	1.000	0.991 **
		Sig. (2-tailed)	.	0.000
		N	7	7
	Pop	Correlation Coefficient	0.991 **	1.000
		Sig. (2-tailed)	0.000	.
		N	7	7

** Correlation is significant at the 0.01 level (2-tailed).

## Data Availability

Not applicable.

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
