# Peer review of "Prevalence and Predictors of COVID-19 Vaccination Acceptance among Greek Health Care Workers and Administrative Officers of Primary Health Care Centers: A Nationwide Study Indicating Aspects for a Role Model"

_vaccines, 2022, doi:10.3390/vaccines10050765_

Round 1

Reviewer 1 Report

The information presented is helpful, but there are a number of areas in which the paper could be improved:

The questionnaire as such is not a validated instrument, and contains questions that are not of obvious relevance as to whether the participants chose to be vaccinated, eg questions 14-17, 22b,c.The most important findings are those presented in Figure 1.

As discussed by the authors, the response rate was poor, perhaps in part due to the complexity of the questionnaire. Also, the focus of the questionnaire was on health care providers and related personnel, and did not include the general population; nor were individuals in the Greek islands included.

Author Response

RESPONSE TO EDITOR AND TO REVIEWERS

Dear Editor,

We are delighted to submit the revised version of the manuscript (ID: vaccines-1684568) to be considered for publication to Vaccines Journal. We provide below a point-by-point response to the comments received by the editor and by the reviewers.

With kind regards,

Varvara Mouchtouri

RESPONSE TO REVIEWER 1

Point 1: The questionnaire as such is not a validated instrument, and contains questions that are not of obvious relevance as to whether the participants chose to be vaccinated, eg questions 14-17, 22b,c.The most important findings are those presented in Figure 1.

We thank the reviewer for giving us the opportunity to elaborate in regard to the validation process of the questionnaire. The validation process is explained as follows:

“The study was supervised by an expert team comprised of an epidemiologist, an occupational health professional and a public health specialist who were responsible for designing and supervising the questionnaire validity (face and content validation). Pilot testing of the first draft questionnaire was conducted to evaluate the time required for its completion, to appraise the clarity of questions addressed to professionals from various backgrounds, and to test the online tool functionality. A total of 20 PHCC professionals, including HCWs and AOs completed the draft questionnaire, which was modified appropriately to produce the final version of the survey. Questionnaires completed during pilot testing were excluded from the final analysis. Internal consistency and reliability of the questionnaire was assessed by estimating Cronbach’s Alpha value of 0.70, which was considered as acceptable. “

Moreover, we would like to clarify that the purpose of the study was both to appraise vaccination intend and practices, and to assess knowledge attitudes and practices relevant to vaccination. In order to investigate whether COVID-19 vaccination hesitant professionals were less acceptant to vaccination in general, the survey questionnaire included questions related to knowledge, attitudes and practices regarding vaccination in general (questions 14-17).   Questions related to COVID-19 vaccines knowledge specifically (22b, c) were useful in order to identify if personnel were well informed about COVID-19 vaccines, if correct information was easily available to them, if this information can create the correct knowledge, and the impact of this knowledge to the COVID vaccine acceptance. To clarify this, we have added a sentence to the methods section of the manuscript.

“In order to identify whether COVID-19 vaccination hesitant professionals were less acceptant to vaccination generally or not, the survey questionnaire included questions related to knowledge, attitudes and practices regarding general vaccination (questions 14-17) and questions related to COVID-19 vaccines specifically ( question 22).”

Point 2. As discussed by the authors, the response rate was poor, perhaps in part due to the complexity of the questionnaire. Also, the focus of the questionnaire was on health care providers and related personnel, and did not include the general population; nor were individuals in the Greek islands included

We thank the reviewer for giving us the opportunity to clarify the target population. As mentioned in the introduction the target population of the study was professionals working in public primary care facilities in Greece, mainland and Greek islands (line 96). The target population can play an important role in the Greek COVID-19 vaccination strategy in providing both information and motivation (through the opinion leaders), as well as administration of vaccines to the Greek population. Therefore, one of the objectives of our study was to assess opinion and practice of personnel of primary health care units towards COVID-19 vaccines (Line 83).

RESPONSE TO REVIEWER 2 COMMENTS

Point 1. In discussion, would include more comment on how this survey was in early 2021 and what survey results now one year later could look like given what is known about the vaccines after a year plus of availability

We thank the reviewer for this important comment, we have added appropriate sentences as suggested to the introduction and discussion sections:

“In late July 2021, the Greek government introduced legislation on mandatory COVID-19 vaccination of personnel of health care facilities in both public and private sectors. However, up to April 2022, the vaccination coverage of HCW according to ECDC COVID-19 vaccines tracker, reached 90.7%, despite the legislation requirement for mandatory vaccination (15).Approximately 9% of HCW preferred to abandon their job or to get   unpaid leave for several months, instead of getting the COVID-19 vaccine. Data related to COVID-19 vaccine acceptance/hesitancy would be important, in order to understand the profile of hesitant professionals and adapting vaccination policies accordingly. “  

“During the study was expected to play a major role in promoting COVID-19 vaccination to the general population and data concerning their intention to get vaccinated and factors related to their decision were important to be investigated when designing and implementing the vaccination strategy. Vaccination monitoring among health care personnel have shown that approximately 9% of HCW refused the COVID-19 vaccination (15). Unfortunately, no follow up study was possible by our team and no relative data are available besides the formal information from the ECDC COVID-19 vaccines tracker referred to the total vaccination coverage among HCW in Greece, which shows small increase of the vaccination coverage compared to our results (90.7% and 85,3% respectively), despite the introduction of legislation for COVID-19 mandatory vaccination of HCW since late July 2021 (15). An ongoing international survey coordinated by WHO is expected to give more insights on changes in vaccination coverage over time and identify factors related to vaccine hesitancy (25)

Point 2. Was there follow-up to see how many of those included actually got the vaccine who said they were or weren't going to? Would include mention in the results / discussion if yes and in discussion if not.

We thank the reviewer for raising this important point. It was not possible by our team to perform a follow up study, but another WHO initiative is ongoing and it is expected to give more insights about the study subject. We have added sentences to the discussion as suggested by the reviewer in order to compare the increase in vaccine coverage since the conduct of our study and after the enforcement of legislation for vaccination of health care staff.

“During the study period, vaccination of the general public was  optional. The target group of our study was expected to play a major role in promoting COVID-19 vaccination to the general population and data concerning their intention to get vaccinated and factors related to their decision were important to be investigated when designing and implementing the vaccination strategy. Vaccination monitoring among health care personnel have shown that approximately 9% of HCW refused the COVID-19 vaccination (15). Unfortunately, no follow up study was possible by our team and no relative data are available besides the formal information from the ECDC COVID-19 vaccines tracker referred to the total vaccination coverage among HCW in Greece, which shows small increase of the vaccination coverage compared to our results (90.7% and 85,3% respectively), despite the introduction of legislation for COVID-19 mandatory vaccination of HCW since late July 2021 (15). An ongoing international survey coordinated by WHO is expected to give more insights on changes in vaccination coverage over time and identify factors related to vaccine hesitancy (25)

Point 3.Would suggest some results/discussion or supplemental tables for the table 2 answers based on the table 1 characteristics. 

We thank the reviewer for this suggestion. We have added a paragraph to the results  section as suggested.

“Variables associated with general vaccination knowledge (Q14) and positive vaccination perceptions(Q15) as well as COVID-19 vaccine knowledge (Q22) are shown in table 7. Generally, top score to the aforementioned questions was positively associate with higher educational level (PR:1.98, 95%CI:1.60-2.46, PR: 2.83, 95%CI:1.96-4.09, PR: 2.52, 95%CI:1.87-3.38) , being physician or health promotion specialist (PR:1.49 95%CI:1.37-1.63, PR: 2.40, 95%CI:2.04-2.82, PR: 1.61, 95%CI:1.44-1.81), being vaccinate for seasonal influenza (PR: 1.26, 95%CI:1.16-1.37, PR1.31, 95%CI:1.23-1.39, PR: 1.24, 95%CI:1.16-1.33), having less concerns about Covid-19 vaccine novelty (PR: 1.3, 95%CI:1.17-1.44, PR: 1.45, 95%CI:1.34-1.57, PR: 1.37, 95%CI:1.25-1.49) and believing in COVID-19 vaccine mandate (PR:1.3, 95%CI:1.17-1.44, PR: 1.37, CI:1.23-1.53, PR: 1.34, 95%CI:1.19-1.51).”

RESPONSE TO REVIEWER 3 COMMENTS

Point 1 Discussion needs more polishing

We thank the reviewer for giving us the opportunity to improve the discussion section. Several paragraphs were added to the discussion part as suggested.

“During the study period, vaccination of both HCW and the general public was optional. The target group of our study was expected to play a major role in promoting COVID-19 vaccination to the general population and data concerning their intention to get vaccinated and factors related to their decision were important to be investigated when designing and implementing the vaccination strategy. Vaccination monitoring among health care personnel have shown that approximately 9% of HCW refused the COVID-19 vaccination as of April 2022 (15). Our study reported 85.3% vaccine acceptance during the study period (February to June 2021), while data available from the ECDC COVID-19 vaccines tracker reported that the total vaccination coverage among HCW in Greece is currently 90.7%, which demonstrate an increase of approximately 5%, after the introduction of legislation for COVID-19 mandatory vaccination of HCW since late July 2021 (15).  Unfortunately, no follow up study was conducted by our team.  An ongoing international survey coordinated by WHO is expected to give more insights on changes in vaccination coverage over time and identify factors related to vaccine hesitancy (25).

 “By the end of April, more than 8 months after introducing legislation for mandatory vaccination of HCW, 9.3% of HCW refused vaccination, despite financial and professional adverse consequences. The results of our survey, depicted that mandatory legislation might not be 100% successful in addressing vaccine hesitancy”

Point 2  The conclusion needs to be more elaborative and should highlight future directions with possible limitations

We thank the reviewer for giving us the opportunity to improve the conclusion section. Additional sentences highlighting the future directions and possible limitations have been added (line 359-367)

“Several cultural and behavioral barriers might play significant role to professionals denying COVID-19 vaccination, which seem to be present even after mandatory measures have been taken. Additional studies in the field could provide useful information and tools to tackle vaccine hesitancy more efficiently. Adequate and constant information/ education about COVID-19 vaccines could be the major tool to increase vaccination coverage of both HCW and general population. Vaccination strategies should consider the reasons that HCW refused COVID-19 vaccines as reported by responders of our study. Addressing fear and providing further information about vaccines through information campaigns for HCW could potentially increase vaccine coverage among HCW”

Point 3 My only concern is that it is an almost a year-old study, and the current situation might be different than what is stated on paper. There should not have been this much delay between collecting data and publishing a survey and making it public. So, the authors should add in the introduction the current figures that received covid-19 vaccinations and discuss the same in the discussion section. At least readers will get an idea of what changed after this survey. 

We agree with the reviewer that we need to give data to the readers about the vaccine coverage the time of the study and currently. We have added those data in both the introduction and the discussion section as suggested.

“In late July 2021, the Greek government introduced legislation on mandatory COVID-19 vaccination of personnel of health care facilities in both public and private sectors. However, up to April 2022, the vaccination coverage of HCW according to ECDC COVID-19 vaccines tracker, reached 90.7%, despite the legislation requirement for mandatory vaccination (15). Approximately 9% of HCW preferred to abandon their job or to get   unpaid leave for several months, instead of getting the COVID-19 vaccine. Data related to COVID-19 vaccine acceptance/hesitancy would be important, in order to understand the profile of hesitant professionals and adapting vaccination policies accordingly. “  ( introduction)

During the study period, vaccination of both HCW and the general public was optional. The target group of our study was expected to play a major role in promoting COVID-19 vaccination to the general population and data concerning their intention to get vaccinated and factors related to their decision were important to be investigated when designing and implementing the vaccination strategy. Vaccination monitoring among health care personnel have shown that approximately 9% of HCW refused the COVID-19 vaccination as of April 2022 (15). Our study reported 85.3% vaccine acceptance during the study period (February to June 2021), while data available from the ECDC COVID-19 vaccines tracker reported that the total vaccination coverage among HCW in Greece is currently 90.7%, which demonstrate an increase of approximately 5%, after the introduction of legislation for COVID-19 mandatory vaccination of HCW since late July 2021 (15).  Unfortunately, no follow up study was conducted by our team.  An ongoing international survey coordinated by WHO is expected to give more insights on changes in vaccination coverage over time and identify factors related to vaccine hesitancy (25).” (Discussion)

By the end of April 2022, more than 8 months after introducing legislation for mandatory vaccination of HCW, 9.3% of HCW continue to refuse   vaccination, despite financial and professional adverse consequences. The results of our survey, depicted that mandatory legislation might not be 100% successful in addressing vaccine hesitancy. (Discussion)

Reviewer 2 Report

This is a well written paper describing the results of a survey conducted in Greece evaluating the uptake/planned uptake of COVID-19 vaccination early in 2021 in health care workers in Greece. 

Minor comments

  1. In discussion, would include more comment on how this survey was in early 2021 and what survey results now one year later could look like given what is known about the vaccines after a year plus of availability
  2.  Was there followup to see how many of those included actually got the vaccine who said they were or weren't going to? Would include mention in the results / discussion if yes and in discussion if not.
  3. Would suggest some results/discussion or supplemental tables for the table 2 answers based on the table 1 characteristics. 

Author Response

(The authors gave the same response as above.)

Reviewer 3 Report

The study underpins the COVID-19 vaccine cover and its acceptance among healthcare workers. There was a high level of acceptance of COVID-19 among PHCC staff in Greece; however, specific subgroups (nurses) were hesitant. In light of the fact that healthcare workers serve as community role models, the authors suggest that more efforts be made to improve the acceptance of the COVID-19 vaccine. 

Overall, the paper is well written. 

-Abstract highlights the importance of the study.

-Introduction is short and confirmed, well connected to the manuscript's contents. 

-Results are fairly well elaborated. 

-Discussion needs more polishing

-Conclusion: The conclusion needs to be more elaborative and should highlight future directions with possible limitations. 

My only concern is that it is an almost a year-old study, and the current situation might be different than what is stated on paper. There should not have been this much delay between collecting data and publishing a survey and making it public. So, the authors should add in the introduction the current figures that received covid-19 vaccinations and discuss the same in the discussion section. At least readers will get an idea of what changed after this survey. 

I believe a high percentage of Europeans still refuse to receive covid vaccinations.

Author Response

(The authors gave the same response as above.)

Round 2

Reviewer 1 Report

The authors' responses are appreciated.